# A multistationary loop model of ALS unveils critical molecular interactions involving mitochondria and glucose metabolism

**Bruno Burlando**[1], **Marco Milanese**[1], **Giulia Giordano**[2,3]\*, **Tiziana Bonifacino**[1], **Silvia Ravera**[4], **Franco Blanchini**[5], **Giambattista Bonanno**[1,6]

**1** Department of Pharmacy, University of Genova, Genova, Italy, **2** Department of Industrial Engineering, University of Trento, Trento, Italy, **3** Delft Center for Systems and Control, Delft University of Technology, Delft, The Netherlands, **4** Department of Experimental Medicine, University of Genova, Genova, Italy, **5** Dipartimento di Scienze Matematiche, Informatiche e Fisiche, University of Udine, Udine, Italy, **6** IRCCS—Ospedale Policlinico San Martino, Genova, Italy

\* g.giordano@tudelft.nl

**Data Availability Statement:** All relevant data are within the manuscript.

**Funding:** Study financially supported by the Italian Ministry of Education, University and Research

## Abstract

Amyotrophic lateral sclerosis (ALS) is a poor-prognosis disease with puzzling pathogenesis and inconclusive treatments. We develop a mathematical model of ALS based on a system of interactive feedback loops, focusing on the mutant SOD1$^{G93A}$ mouse. Misfolded mutant SOD1 aggregates in motor neuron (MN) mitochondria and triggers a first loop characterized by oxidative phosphorylation impairment, AMP kinase over-activation, 6-phosphofructo-2-kinase (PFK3) rise, glucose metabolism shift from pentose phosphate pathway (PPP) to glycolysis, cell redox unbalance, and further worsening of mitochondrial dysfunction. Oxidative stress then triggers a second loop, involving the excitotoxic glutamatergic cascade, with cytosolic Ca$^{2+}$ overload, increase of PFK3 expression, and further metabolic shift from PPP to glycolysis. Finally, cytosolic Ca$^{2+}$ rise is also detrimental to mitochondria and oxidative phosphorylation, thus closing a third loop. These three loops are overlapped and positive (including an even number of inhibitory steps), hence they form a candidate multistationary (bistable) system. To describe the system dynamics, we model the interactions among the functional agents with differential equations. The system turns out to admit two stable equilibria: the healthy state, with high oxidative phosphorylation and preferential PPP, and the pathological state, with AMP kinase activation, PFK3 over expression, oxidative stress, excitotoxicity and MN degeneration. We demonstrate that the loop system is monotone: all functional agents consistently act toward the healthy or pathological condition, depending on low or high mutant SOD1 input. We also highlight that molecular interactions involving PFK3 are crucial, as their deletion disrupts the system's bistability leading to a single healthy equilibrium point. Hence, our mathematical model unveils that promising ALS management strategies should be targeted to mechanisms that keep low PFK3 expression and activity within MNs.

(SIR project n.RBSI14B1Z1) (MM), the Fondazione Compagnia di San Paolo (project n.2018.AAI629. U730/SD/pv) (GB), and the Motor Neurone Disease Association (project n. April16/848-791) (GB). The funders had no role in study design, data collection and analysis, decision to publish, or preparation of the manuscript.

**Competing interests:** The authors have declared that no competing interests exist.

## Introduction

Neurodegenerative diseases are progressive neuronal death syndromes characterized by apoptosis and necrosis processes. In amyotrophic lateral sclerosis (ALS), these events involve upper and lower motor neurons (MNs) at cortical, brainstem, and spinal levels, followed by muscle weakness and paralysis. This syndrome is part of a series of MN diseases that also includes pseudobulbar and progressive bulbar palsy, progressive muscular atrophy, and primary lateral sclerosis [1]. It is a complex and heterogeneous condition lacking established early markers, which makes its diagnosis challenging and essentially based on clinical evidence [2]. The disease has a poor prognosis, patients dying within 3–5 years since diagnosis, and the only approved drugs are the anti-excitotoxic riluzole and the free-radical scavenger edaravone [3].

Two different forms of ALS are known, familial and sporadic, with the former representing about 5–10% of total cases [4]. A number of genes potentially involved in familial ALS have been identified, with about 25% of these forms being due to mutations in the gene encoding for Cu/Zn superoxide dismutase type 1 (SOD1). Among these latter, the most abundant one is the glycine substitution with alanine at position 93 (G93A). Following this kind of evidence, a mouse strain expressing SOD1$^{G93A}$ protein has been developed and widely adopted as an experimental animal model in ALS molecular and preclinical studies [5].

A number of molecular mechanisms concerning MN degeneration in ALS have been described, among which major topics include glutamate excitotoxicity, structural and functional disorders of mitochondria, impaired axonal functions, protein misfolding linked to endoplasmic reticulum stress, and oxidative stress [6–10]. Also, the involvement of cell types other than MNs in the onset and progression of the disease has been documented, mostly astrocytes, microglia, and oligodendrocytes, leading to the concept of non-cell autonomous pathogenesis [11–13]. However, it is generally assumed that MNs are the site of disease onset, while glial cells are believed to influence the disease by shifting from a homeostatic role to aberrant reactivity that speeds up neuronal degeneration [14, 15].

The main hindrance to the development of effective treatments for ALS is the poor knowledge of pathogenetic mechanisms. Despite a bulk of data collected through years at the molecular, cellular, and organism levels, on both animal models and human subjects, and a number of hypotheses put forward about pathogenesis, the identification of a *bona fide* primary event for the onset of the disease is still lacking, thus making pharmacological and clinical strategies disappointedly weak [16]. By considering previous published data from our laboratory [17, 18], as well as, at present, unpublished data, and combining them with other inputs from literature, in this study we propose a model based on interactive feedback loops mainly focused on the SOD1$^{G93A}$ mouse model. In particular, we envisage a feedback loop interaction between mitochondria and glucose metabolism in MNs. Such a model provides hints for a description of ALS insurgence in terms of a multistationary system driven by positive loops, i.e. chains of interactions that are overall activating, which undergoes transitions among different equilibrium points or steady-state configurations [19].

## The model

### Biological background

We started from the obvious consideration that mutant SOD1 protein must have an essential role in pathogenesis. A clue for an interpretation of this role is offered by functional and pro-apoptotic degeneration of mitochondria observed in spinal cord MNs, starting from presymptomatic stages of the disease [17, 20]. These data can be linked to the finding that ALS

mutant SOD1 forms aggregates of misfolded units specifically on the outer membrane of spinal MN mitochondria, eventually impairing mitochondrial function [21, 22]. This event can play a seminal role in the onset of the disease, considering that in the central nervous system (CNS) the energetic metabolism is distinctly partitioned, with astrocytes being more glycolytic and neurons more oxidative, involving higher energy metabolism via oxidative phosphorylation (OXPHOS) in neurons. Moreover, neurons also display a high rate of the pentose phosphate pathway (PPP), while astrocytes are known to provide neurons with lactate that is then converted to pyruvate and contributes to satisfy the needs for mitochondrial metabolism [23, 24]. Given these premises, another important piece of evidence for the building of our model is the apparent inversion of MN energetic metabolism in SOD1[G93A] mice, from prevalently oxidative under healthy conditions to prevalently glycolytic during pathogenesis [17, 18]. Moreover, as observed for mutant SOD1 binding to mitochondria, also the metabolism inversion occurs in the spinal cord at an early stage of the disease and then appears in the motor cortex at a later symptomatic stage [18].

Consistent with the mitochondrial impairment induced by mutant SOD1, a decreased ATP/AMP ratio has been found in spinal cord MNs in SOD1[G93A] mice already at pre-symptomatic stages [17]. As known, the AMP accumulation promotes the activity of the energy sensor AMP-activated kinase (AMPK) [25] and, consistently, abnormally upregulated AMPK has been reported for both SOD1[G93A] mice and ALS patients [26]. Moreover, the antidiabetic drug metformin, which is known to activate AMPK [27], has been found to accelerate both symptom onset and disease progression in SOD1[G93A] mice [28]. As a consequence, these data lead directly to glucose catabolism as a major player in the disease.

Glycolysis is known to be controlled by the enzyme 6-phosphofructo-1-kinase, which is regulated allosterically by fructose-2,6-bisphosphate. In the brain, the biosynthesis of the latter is operated almost exclusively by 6-phosphofructo-2-kinase/fructose-2,6-bisphosphatase-3 (PFK3) that is activated through phosphorylation by AMPK [29]. In normal neurons, glycolysis is low, due to limited PFK3 activity, in favor of an elevated PPP rate [24]. Hence, it could be argued that in MNs developing ALS an increase of AMPK activity would lead to PFK3 activation, thereby producing a shift of glucose catabolism, from PPP to glycolysis [23]. Now, being PPP the major producer of reducing equivalents in the cell, such a shift would be detrimental for the cell redox balance, thus exacerbating an already compromised scenario characterized by mitochondrial dysfunction and the correlated increase of oxidative stress. In fact, mitochondria are major cellular sites of reactive oxygen species (ROS) production, and this activity abnormally increases upon dysfunction of the electron transport chain, as observed in spinal cord synaptic nerve terminals of SOD1[G93A] mice [17]. However, PFK3 is known to be expressed in neurons at low levels due to its efficient targeting to proteasomal degradation by the E3 ubiquitin ligase anaphase-promoting complex-Cdh1 (APC[Cdh1]) [30]. This could slow down the AMPK-dependent PFK3 activation, but as we will see the SOD1-induced mitochondrial impairment is also likely to promote PFK3 expression.

A rise in ROS production is known to affect the equilibrium of the glutamatergic tripartite synapses, where astrocytes are known to play an essential role in glutamate removal via EAAT transporters [31, 32]. Oxidative stress has been shown to affect glutamate clearance from the synaptic space, thus paving the way to excitotoxic effects mediated by excessive $Ca^{2+}$ entry into MNs [33]. It has been shown that one of the consequences of excessive $Ca^{2+}$ entry is a tissue-selective aberrant activation of the calpain/calpastatin system [34, 35], which, among its several targets, leads to cleavage of cyclin-dependent kinase 5 (CDK5) activator p35 into membrane-bound p10 and cytosolic p25 subunits [36]. The p25 subunit forms a complex with CDK5 that promotes a deregulated activation of the kinase itself [37], and it has been shown that the CDK5-p25 complex hyper-phosphorylates Cdh1, thus inhibiting the formation of the APC[Cdh1]

complex [38, 39]. This has been confirmed in cortical neurons, showing that simulation of excitotoxicity through N-methyl-D-aspartate receptor activation inhibits CDK5/Cdh1 and increases cytosolic PFK3 levels, thereby switching the glucose catabolism from PPP to glycolysis [23]. Finally, the enhanced cytosolic $Ca^{2+}$, induced by excessive glutamate, has been found to trigger also mitochondrial $Ca^{2+}$ overload with increased ROS production [40, 41].

Starting from the large amount of evidence described above, we derived a system of three functional loops strictly intertwined, and developed a mathematical model able to identify the key hub of the three loops and the related most relevant and critical targets for advanced therapeutic solutions in ALS.

## Loop system

**Loop 1.** According to the above evidence, by affecting spinal MN mitochondria, mutant SOD1 would have the effect of increasing AMPK, activating PFK3, depressing PPP, and reducing the ability of the cell to counteract oxidative stress, thus leading to further mitochondrial damage. This triggers a first positive loop that can be schematically represented by an even number of inhibitory steps and is therefore overall activating: OXPHOS-AMPK-PFK3-PPP-OXPHOS (Fig 1A and 1B). This scenario is intriguing because the dynamics of a positive loop can account for a transition between health and disease [19], while it is also consistent with the well-known role of oxidative stress in the development of ALS [42, 43].

**Loop 2.** As described above, data in the literature indicate that $APC^{Cdh1}$ plays a key role in the regulation of neuronal glucose metabolism by keeping low PFK3 expression, while $APC^{Cdh1}$ inhibition is expected to produce the opposite effect. This biological pathway can be connected to part of Loop 1 thereby closing a second loop, PFK3-PPP-GLUTAMA-TE-$APC^{Cdh1}$-PFK3 (Fig 1A and 1C), containing four inhibitory steps, and therefore being overall positive or activating. This loop is partially overlapped with the previous one, while both loops converge on PFK3, one promoting its expression level, and the other its phosphorylation. Hence, the two loops can act synergistically by strengthening each other and are expected to strongly induce the transition from low glycolysis/high PPP, maintaining redox balance, to high glycolysis/low PPP, leading to redox unbalance and oxidative stress.

**Loop 3.** As excitotoxicity rise, a third positive loop can be envisaged: OXPHO-S-AMPK-PFK3-PPP-GLUTAMATE-OXPHOS (Fig 1A and 1D), also partially overlapped with the previous ones. The abnormal glutamatergic signal triggers an aberrant cytosolic $Ca^{2+}$ rise in spinal cord motoneurons, with a consequent mitochondria $Ca^{2+}$ overload and a dramatic disruption of the mitochondrial membrane potential ($\Delta\Psi m$). The mitochondrial membrane potential ($\Delta\Psi m$) is an essential component in the process of energy storage (ATP) during oxidative phosphorylation. The occurrence of inner membrane disruption significantly compromises the magnitude of $\Delta\Psi m$, thus mitochondrial respiratory chain becomes a significant producer of reactive oxygen species (ROS) [44]. The given excessive production of ROS is again overlapped with the previous two loops and fosters the already compromised pathological scenario.

In summary, the herein-described ALS pathogenesis model involves three overlapped positive loops forming a synergistic system that fuels oxidative stress and excitotoxicity, i.e. two leitmotifs of the disease at the cellular and molecular levels.

## Mathematical model and methods

The dynamics of the loop system (Fig 1) can be studied using mathematical tools from Systems and Control Theory. We propose a system of differential equations, modeling the dynamical interactions between pairs of functional agents. We assume that each functional agent, say $x_i$,

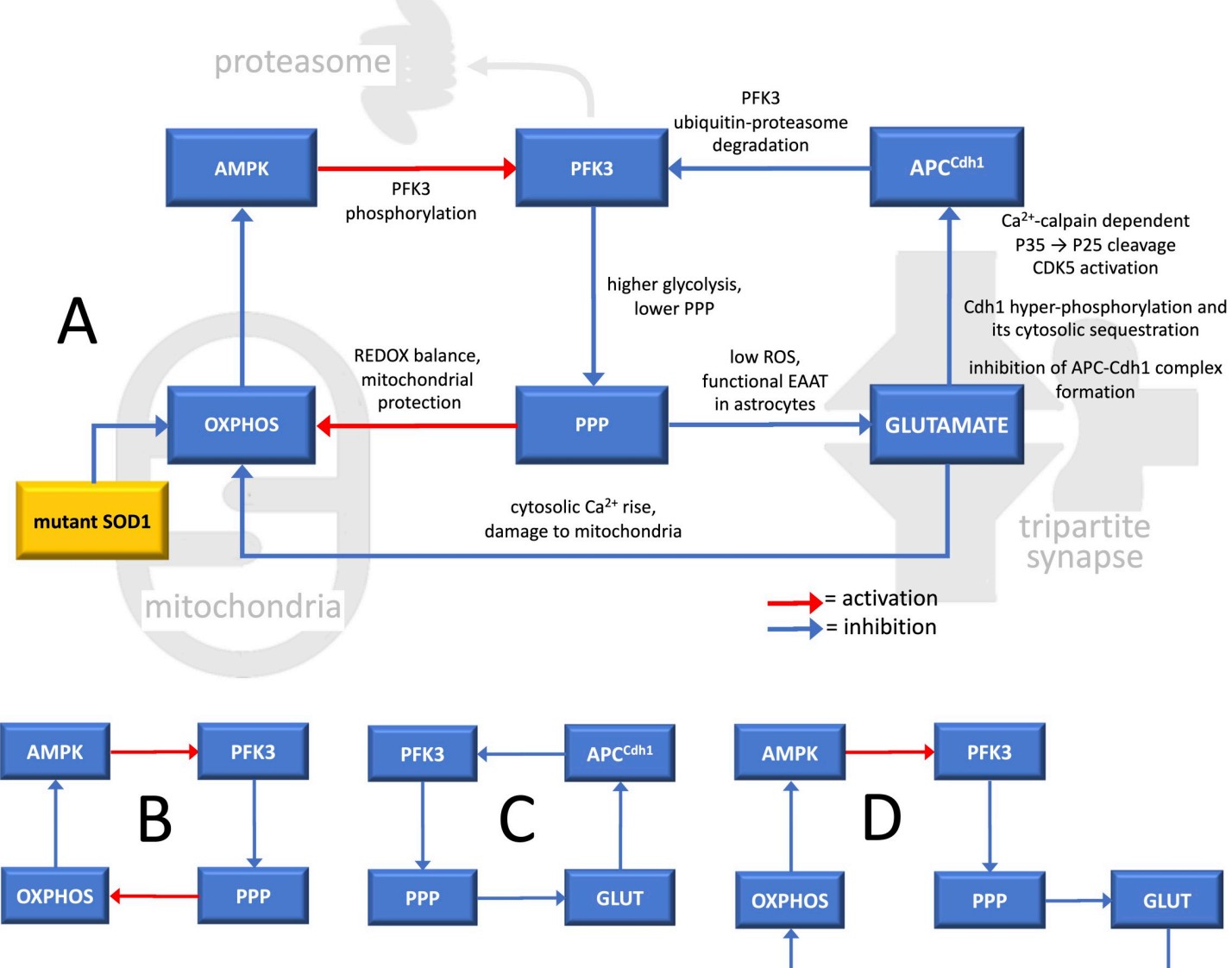

**Fig 1. Diagrams of the feedback loop system involved in ALS pathogenesis in spinal cord motor neurons.** (A) Overall diagram of the feedback loop system. Boxes represent functional agents and arcs the effects that each agent exerts on another one. Mutant SOD1 is the triggering element that, by affecting mitochondria, induces a dynamical transition in the loop system thereby causing the insurgence of disease. (B, C, D) Diagrams of the three single loops constituting the loop system. AMPK: AMP-activated kinase; APC$^{Cdh1}$: E3 ubiquitin ligase anaphase-promoting complex-Cdh1; OXPHOS: oxidative phosphorylation; PFK3: 6-phosphofructo-2-kinase/fructose-2,6-bisphosphatase-3; PPP: pentose phosphate pathway.

is subject to a spontaneous degradation and evolves with time constant $\tau_{x_i}$. We denote by $f$ an activation function, which is monotonically increasing in its argument, and by $h$ an inhibition function, which is monotonically decreasing in its argument. The differential equation describing the evolution of the generic functional agent $x_i$ has therefore the form:

$$\tau_{x_i}\dot{x}_i + x_i = f(x_j) + h(x_k) + \cdots$$

where $x_j$ is a functional agent that activates $x_i$, while $x_k$ is a functional agent that inhibits $x_i$. The

system Jacobian matrix $J$ is then the square matrix whose $(i, k)$ entry can be computed as

$$J_{ik} = \frac{\partial \dot{x}_i}{\partial x_k}$$

Since the generic functional agent $x_k$ either activates $x_i$ or inhibits $x_i$, or has no influence on $x_i$, the corresponding entry of $J$ is sign determined: $\text{sign}(J_{ik}) \in \{+,-,0\}$.

Common examples of the involved monotonic functions are the well-known Hill functions [45, 46], having the following expressions:

$$f(x) = \frac{\alpha x^p}{1 + \beta x^p} \qquad \text{and} \qquad h(x) = \frac{\gamma}{1 + \delta x^p}$$

for increasing and decreasing functions, respectively, where $p$ is the Hill coefficient and the Greek letters represent positive real parameters. However, the results we provide here exclusively depend on the qualitative monotonicity property and do not rely on any assumed exact functional expressions for activations and inhibitions, or on the value of any involved parameter.

To describe the dynamic evolution of the system we denote the concentrations or activities of the functional agents as follows:

- [mutant SOD1] = $u$,

- [mitochondrial OXPHOS] = $m$,

- [AMPK] = $a$,

- [PFK3] = $k$,

- [APC$^{\text{Cdh1}}$] = $c$,

- [glutamate] = $g$,

- [PPP] = $p$

resulting in the simplified scheme in Fig 2.

Then, the dynamics associated with the feedback loop arrangement (Fig 2) is described by the following system of ordinary differential equations, where the activating and inhibitory interactions visualized in Figs 1 and 2 are modelled in terms of monotonic functions:

$$\tau_a \dot{a} + a = h_6(m) \tag{1}$$

$$\tau_k \dot{k} + k = f_1(a) + h_4(c) \tag{2}$$

$$\tau_c \dot{c} + c = h_2(g) \tag{3}$$

$$\tau_m \dot{m} + m = h_7(u) + h_1(g) + f_2(p) \tag{4}$$

$$\tau_p \dot{p} + p = h_5(k) \tag{5}$$

$$\tau_g \dot{g} + g = h_3(p) \tag{6}$$

The dynamic behavior of the loop system can then be qualitatively determined by associating the equation system with its interaction matrix $S$, which can be computed as the sign pattern of the system Jacobian matrix $J$: $S = \text{sign}(J)$, where the sign function for matrices is defined elementwise. In view of the monotonicity of the involved interaction functions, the

interaction matrix $S$ can be derived exclusively based on the qualitative knowledge about the system that is summarized in the graphs in Figs 1 and 2, while no information on the involved functional expressions and parameters is needed. Then, the sign of the loops present in the graph associated with the signed matrix $S$ can provide useful insight into the system's behavior: as shown in [19], if all the loops are positive, then the system is a candidate multistationary system, hence it can admit multiple stable equilibria corresponding to different configurations.

A functional agent can influence another not only directly, but also indirectly. For instance, in Fig 2 we see that $p$ activates $m$ directly, but it also inhibits $g$ that in turn inhibits $m$. How can we compute a net interaction matrix, which takes into account the resulting effect of all possible direct and indirect influences between pairs of functional agents? To display the overall net signed effect of each functional agent on each of the others, when the system is perturbed around a stable equilibrium point, we can compute the structural influence matrix according to the methods proposed in [47]. The structural influence matrix can be computed in closed form as the sign pattern of the adjoint of the negative Jacobian matrix, sign[adj(−J)]; equivalently, when the Jacobian is non-singular, the $(i, k)$ entry of the structural influence matrix can be computed as

$$\text{sign}\left(\frac{\det\begin{bmatrix} -J & -C_k \\ R_i & 0 \end{bmatrix}}{\det(-J)}\right)$$

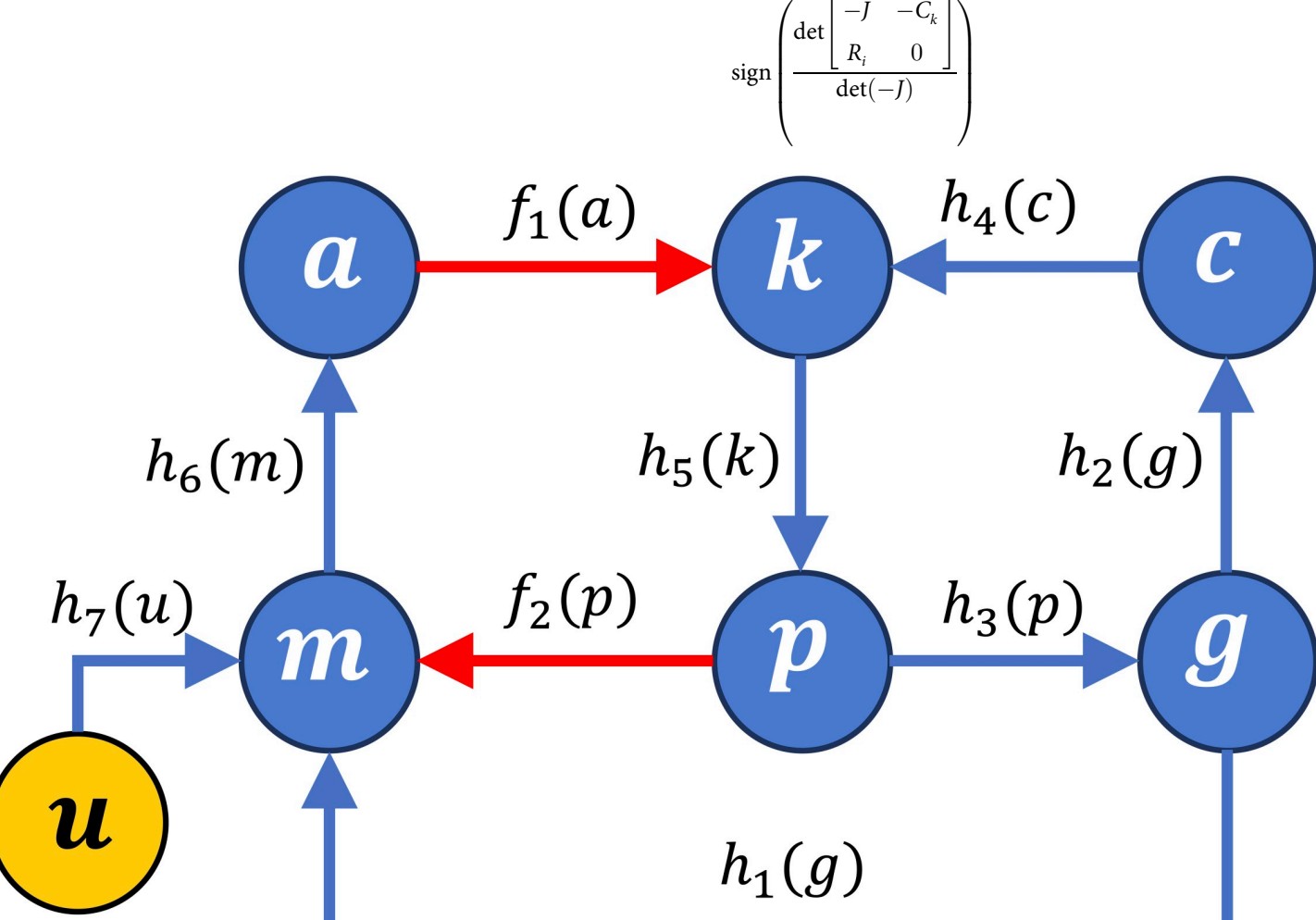

**Fig 2. Graph associated with the system of differential Eqs (1)-(6).** The functional agents shown in Fig 1 are denoted as [mutant SOD1] $=u$, [OXPHOS] $= m$, [AMPK] $= a$, [PFK3] $= k$, [APC$^{\text{Cdh1}}$] $= c$, [GLUTAMATE] $= g$, and [PPP] $= p$, while their interactions are associated with activation functions $f$ and inhibition functions $h$. Arcs as in Fig 1.

where $R_i$ is a row vector that has a one entry in position $i$ and is zero elsewhere, while $C_k$ is a column vector that has a one entry in position $k$ and is zero elsewhere. Since we assume stability of the equilibrium, we have $\det(-J) > 0$, hence we can equivalently consider

$$\text{sign}\left( \det \begin{bmatrix} -J & -C_k \\ R_i & 0 \end{bmatrix} \right).$$

Each entry of the structural influence matrix can be either '+', if the net overall influence is always positive regardless of the system parameters; '−', if the net overall influence is always negative regardless of the system parameters; '0' if the net overall influence is always zero regardless of the system parameters; or '?' if the net overall influence can have a different sign depending on the system parameters (for instance, this happens if functional agent A directly inhibits functional agent C, but at the same time activates functional agent B that in turn activates C: then, we have two contrasting effects, a direct inhibition and an indirect activation, and depending on the numerical strength of the interactions the overall net effect can be either positive or negative). Efficient approaches for computing the structural influence matrix are proposed in [47] for a vast class of systems, including (1)-(6).

## Results

The system (1)-(6), which describes the dynamic evolution of the functional agents involved in ALS pathogenesis, can be associated with two signed matrices, as discussed in the "Mathematical model and methods" section. The first is the interaction matrix $S$, shown in Equation (7), which displays the direct influence of each functional agent on each of the others. Matrix $S$ includes the signs of the entries of the system Jacobian matrix (activation functions lead to a '+' sign, while inhibition functions lead to a '−', and no interaction leads to a '0'):

|  | Direct influence of: | AMPK | PFK3 | APC$^{Cdh1}$ | OXPHOS | PPP | GLUTAMATE |  |
|---|---|---|---|---|---|---|---|---|
|  | on AMPK | − | 0 | 0 | − | 0 | 0 |  |
|  | on PFK3 | + | − | − | 0 | 0 | 0 |  |
| $S =$ | on APC$^{Cdh1}$ | 0 | 0 | − | 0 | 0 | − | (7) |
|  | on OXPHOS | 0 | 0 | 0 | − | + | − |  |
|  | on PPP | 0 | − | 0 | 0 | − | 0 |  |
|  | on GLUTAMATE | 0 | 0 | 0 | 0 | − | − |  |

In addition, $S_u$ is the signed input matrix (8), which includes the signs of the derivatives of the system equations with respect to the "triggering input" $u$ (mutant SOD1):

|  | Direct influence of: | mutant SOD1 |  |
|---|---|---|---|
|  | on AMPK | 0 |  |
|  | on PFK3 | 0 |  |
| $S_u =$ | on APC$^{Cdh1}$ | 0 | (8) |
|  | on OXPHOS | − |  |
|  | on PPP | 0 |  |
|  | on GLUTAMATE | 0 |  |

As already observed, three loops are present in the system and they are all positive, because include an even number of inhibitory (i.e., negative) interactions. These loops can be easily

visualized by inspecting matrix $S$ (7) as well as the graph in Figs 1 and 2. The first loop is $m-a-k-p-m$ (Fig 1B), the second is $k-p-g-c-k$ (Fig 1C), and the third is $m-a-k-p-g-m$ (Fig 1D).

Because only positive loops are present in the associated graph, this is a candidate multistationary (bistable) system according to the results in [19]. As a consequence, its Jacobian matrix has a dominant eigenvalue that is real; this entails that an equilibrium can become unstable exclusively due to a real eigenvalue that crosses the imaginary axis, turning from negative to positive. This type of instability is typically associated with a bifurcation leading to the destabilization of one equilibrium and the concurrent appearance of additional stable equilibria. In our system, among the stable equilibria one corresponds to "healthy condition" and another to "pathological condition", explaining the possible onset of the disease.

Since all the loops are positive, we can show that the system in Eqs (1)–(6) is a monotone system, i.e. its Jacobian matrix can be turned into a Metzler matrix (i.e., a matrix with nonnegative off-diagonal entries) by changing the sign of some variables. In fact, changing the sign of the variables $a$, $k$, and $g$ leads to the new signed matrices:

|  | Direct influence of: | −AMPK | -PFK3 | APC$^{Cdh1}$ | OXPHOS | PPP | −GLUTAMATE |  |
|---|---|---|---|---|---|---|---|---|
|  | on −AMPK | − | 0 | 0 | + | 0 | 0 |  |
|  | on −PFK3 | + | − | + | 0 | 0 | 0 |  |
| $\hat{S} =$ | on APC$^{Cdh1}$ | 0 | 0 | − | 0 | 0 | + | (9) |
|  | on OXPHOS | 0 | 0 | 0 | − | + | + |  |
|  | on PPP | 0 | + | 0 | 0 | − | 0 |  |
|  | on −GLUTAMATE | 0 | 0 | 0 | 0 | + | − |  |

and

|  |  | Direct influence of: mutant SOD1 |  |  |
|---|---|---|---|---|
|  |  | on −AMPK | 0 |  |
|  |  | on −PFK3 | 0 |  |
| $\hat{S}_u =$ |  | on APC$^{Cdh1}$ | 0 | (10) |
|  |  | on OXPHOS | − |  |
|  |  | on PPP | 0 |  |
|  |  | on −GLUTAMATE | 0 |  |

where clearly $\hat{S}$ (9) is a Metzler matrix (all its off-diagonal entries are nonnegative). Being the system monotone, oscillatory instability can be ruled out: either the system has a single stable equilibrium point, or one equilibrium can be unstable in the presence of two coexisting stable equilibria (e.g. one "healthy" and one "pathological").

In particular, we can think of a scenario with two qualitative stable equilibria: if $u$ (mutant SOD1) is at a high level, then it contributes to keep $m$ (OXPHOS) low, which keeps $a$ (AMPK) high, which keeps $k$ (PFK3) high as well, which keeps $p$ (PPP) low (which contributes to keep $m$ low, consistently), and this keeps $g$ (glutamate) high (which also consistently keeps $m$ low), which keeps $c$ (APC$^{Cdh1}$) low, which consistently keeps $k$ high. On the other hand, if $u$ is at a low level, then it contributes to keep $m$ high, which keeps $a$ low, which keeps $k$ low as well, which keeps $p$ high (which consistently contributes to keep $m$ high), and this keeps $g$ low (which also consistently keeps $m$ high), which keeps $c$ high, which consistently keeps $k$ low.

Both equilibria are stable (all the activations and inhibitions are consistent, so that there cannot be oscillations, which could only happen in the presence of negative feedback loops, i.e.

loops with an odd number of inhibitions, as shown in [19]) and a variation in the input $u$, which is associated with mutant SOD1, can induce a transition from one stable steady state to the other.

We can assess the effect of an input perturbation on the steady state of the system, and the ensuing steady-state variation, with the qualitative, parameter-free approach for the computation of structural influence matrices previously outlined in [47]. In particular, if we assume stability of the equilibrium at which the system is resting, the effect of a sudden increase in the value of $u$ on the value of the variables $(a, k, c, m, p, g)$ can be captured by the structural input-output influence matrix:

| Overall influence of: mutant SOD1 | | |
|---|---|---|
| on AMPK | + | |
| on PFK3 | + | |
| on APC$^{Cdh1}$ | − | (11) |
| on OXPHOS | − | |
| on PPP | − | |
| on GLUTAMATE | + | |

where the '+' entries mean that the new steady-state value of variables $a$, $k$ and $g$ always increases, while the '−' entries mean that the new steady-state value of variables $c$, $m$ and $p$ always decreases, if the value of $u$ suddenly increases, regardless of the expressions for the activation and inhibition functions and regardless of parameter values.

More in general, we can compute the structural influence matrix (12), whose $(i, k)$ entry expresses the structural sign of the variation in the steady state of the $i$th variable due to the addition of a persistent input acting on the equation of the $k$th variable. For the considered system, if we assume stability of the equilibrium, the structural influence matrix turns out to be:

| Overall influence of: | AMPK | PFK3 | APC$^{Cdh1}$ | OXPHOS | PPP | GLUTAMATE | |
|---|---|---|---|---|---|---|---|
| on AMPK | + | + | − | − | − | + | |
| on PFK3 | + | + | − | − | − | + | |
| on APC$^{Cdh1}$ | − | − | + | + | + | − | (12) |
| on OXPHOS | − | − | + | + | + | − | |
| on PPP | − | − | + | + | + | − | |
| on GLUTAMATE | + | + | − | − | − | + | |

In this case, computing the structural influence matrix is straightforward if we notice that the Jacobian matrix $S$ (7) can be turned into a Metzler matrix by changing the sign of the first, second and sixth variable. In fact, as proven in [47], for a stable Metzler matrix that is also irreducible, as is the case for the Metzler matrix $\hat{S}$ (9), the structural influence matrix has all positive entries. Then, we can obtain the structural influence matrix for our system by considering a fully positive sign matrix and changing sign to the first, second and sixth row and column, which leads to the structural influence matrix (12) shown above.

Differently from the signed interaction matrix $S$, which is associated with the sign pattern of the Jacobian matrix and therefore expresses the direct signed influence of each functional agent on each of the others (so that the matrix entries correspond to the inhibiting/activating arcs in Figs 1 and 2), the influence matrix captures the overall net signed influence, which results from the combination of direct and several indirect effects through different entangled

loops and paths. Note that the fourth column of the structural influence matrix (12) corresponds to the negative of the structural input-output influence matrix (11), exactly because input $u$ acts, with a negative (inhibiting) direct influence, upon the fourth variable, $m$. Interestingly, we can notice that all the net overall influences among functional agents in the considered system are structurally signed, regardless of parameter values.

The achievement of our aim is demonstrated by the following clues, originating from the mathematical model: all the three loops in the system share precisely one interaction, which is the inhibition of $p$ due to $k$, $h_5(k)$. If this single interaction is knocked out, then there are no more loops in the system, and in this case there is a single equilibrium, which must be stable. In fact, by inspecting system (1)-(6) we can notice that, if $h_5(k) = 0$, then, for a given value of the input $\bar{u}$, the values of the variables at steady state can be directly computed as $\bar{p} = 0$, $\bar{g} = h_3(0)$, $\bar{c} = h_2(h_3(0))$, $\bar{m} = h_7(\bar{u}) + h_1(h_3(0)) + f_2(0)$, $\bar{a} = h_6(\bar{m})$ and $\bar{k} = f_1(\bar{a}) + h_4(\bar{c})$, hence a single equilibrium is possible. This highlights the importance of the inhibitory interaction from PFK3 to PPP, which looks crucial to allow bistability, hence the possibility of having both a healthy stable state and a pathological stable state, depending on the initial conditions and on the input $u$ due to mutant SOD1.

Bistability is possible after crossing a bifurcation point due to the variation of a bifurcation parameter (see Fig 3), which in our loop system can be identified with the strength of the interaction between PFK3 and PPP, i.e. the key interaction of the system, corresponding to the parameter $h_5(k)$ in the differential equations. At the beginning of the pathogenic process, the disturbing action of misfolded SOD1 on mitochondria with AMPK activation is followed by a phosphorylation of PFK3 that modifies its enzymatic kinetics [29, 48]. Hence, even if PFK3 is still expressed at low levels, its phosphorylation strengthens the negative interaction between PFK3 and PPP. We assume that this is the event that drives the system to undergo a bifurcation and exhibit bistability (Fig 3). Then, the continuous disturbing input of SOD1, and the consistent monotone action of the functional loop agents, can drive the system to fall onto the stable equilibrium point that represents the pathological condition (Fig 3). Thereafter, if the action of SOD1 further strengthens the key interaction corresponding to $h_5(k)$, the system could cross a second, inversely oriented bifurcation point, hence leaving the bistability zone and remaining positioned at the unique "pathological" equilibrium point (Fig 3). This bifurcation diagram can be achieved by assuming function $h_5(k)$ of the form

$$h_5(k) = \mu h(k),$$

where $h$ is a given decreasing function and $\mu$ is a positive strength parameter. Due to monotonicity, the value of PFK3 in the stable equilibria (the green ones in Fig 3) is an increasing function of $\mu$.

## Discussion

Our pathogenesis model of ALS follows a completely new approach based on Systems and Control Theory. The model embodies some of the most accredited driving factors of the pathology, including mitochondria, oxidative stress and excitotoxicity. The main novelty is that these factors are entangled in a system of feedback loops according to their mutual interactions. The onset and progression of the disease is explained as a transition from one equilibrium point (health) to another one (disease), which can occur in positive loop systems upon the influence of an external stimulus. In our model, constructed on SOD1 mutants, the stimulus is assumed to be a damaging action of misfolded SOD1 proteins on mitochondria, which is also the rationale for adopting the mitochondria/glycolysis loop as the core of the model, among a number of possible etiological anomalies reported in the literature [49].

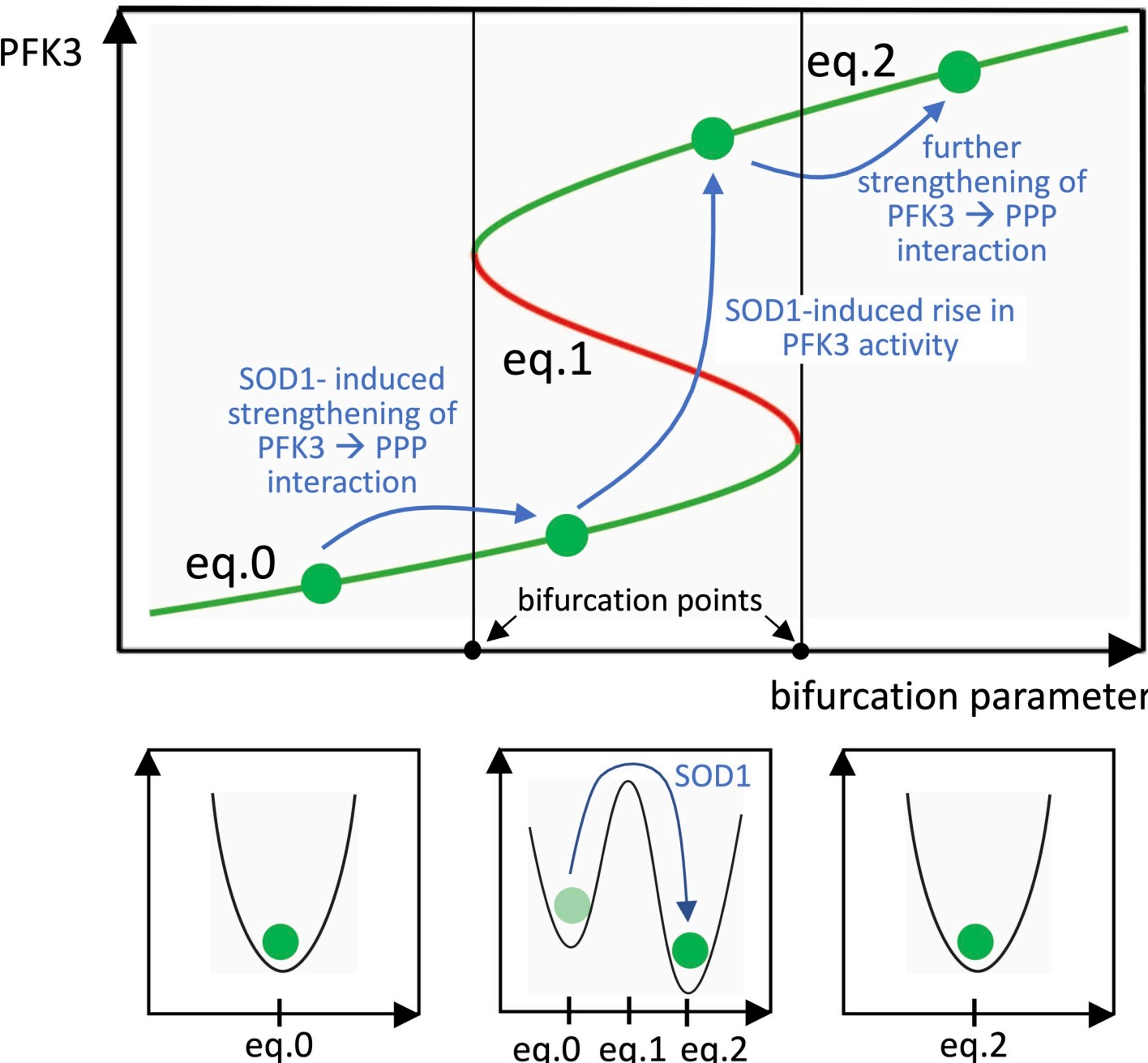

**Fig 3.** Bifurcation diagram (top panel) and corresponding potential energy diagrams (bottom panels). Diagrams represent the evolution of the equilibrium points of the loop system in a phase space consisting of all possible values of PFK3 enzyme activity and of a bifurcation parameter, which is taken as the strength of the interaction between PFK3 and PPP (i.e. term $h_5(k)$ in the differential equations, see text and Fig 2). Due to SOD1-induced AMPK activation, PFK3 is phosphorylated, the interaction strengthens, and the system undergoes a bifurcation and becomes multistationary: two stable equilibrium points (eq.0 and Eq 2, green lines) and an unstable one (Eq 1, red line) coexist. Thereafter, the continuous disturbing action of SOD1 ($h_7(u)$, see Fig 2) induces an increase in PFK3 activity that drives the loop system to the stable equilibrium point (Eq 2) that represents the pathological condition. Finally, a further strengthening of the PFK3/PPP interaction could induce the system to leave the multistationary region, by crossing an inversely oriented bifurcation point, and remain positioned at the unique "pathological" equilibrium point.

The model consists of three positive loops coupled together and acting synergistically, i.e. any loop serves as a sensitization agent for each other, which involves their mutual strengthening and, as generally experienced for these dynamical systems, can lead to the irreversibility of

their transition [50]. We invoke bifurcation dynamics to explain the occurrence of mutistationarity, involving the interaction between mitochondrial-induced alteration of glycolysis/PPP ratio (Fig 1B) with excitotoxicity (Fig 1C and 1D). These events occur in a very early phase of the disease and direct evidence of their relative timing is lacking. However, unbalance of mitochondrial function and enhanced glucose metabolism have been recorded at an early pre-symptomatic stage in spinal cord MNs of SOD1$^{G93A}$ mice [17, 18]. These events possibly open the way to the wide complex of phenomena that have been described in ALS, involving not only MN degeneration, but also astrogliosis, microglia activation and neuroinflammation.

The core of the model is tailored on MNs and bases on data obtained from the SOD1$^{G93A}$ mice model, but it can be extended to other forms of familial ALS, since different SOD1 mutations and other proteins linked to the disease, like TDP-43, FUS, and C9ORF72, have been found to cause mitochondrial defects and disruption of the energy metabolism in MNs [51–53], hence potentially eliciting the same system of loops. Also, the role of the ATP/AMP ratio can explain a hitherto unsolved feature of ALS in SOD1$^{G93A}$ and SOD1$^{G85R}$ mice, i.e. the correlation between neuron size and degeneration, with largest fast-twitch fatigable (FF) MNs being the first to be affected, followed by fast-twitch, fatigue-resistant (FR) MNs, and finally by slow-twitch (S) MNs [54, 55]. FF MNs are the largest ones, and it has been shown that they consume more ATP per action potential to maintain cytosolic Na$^+$ and K$^+$ homeostasis, which makes them more sensitive to ATP depletion caused by mitochondrial dysfunction [56]. Our model provides a prompt explanation for these data, because a more rapid alteration of the ATP/AMP ratio in FF MNs would accelerate the triggering of the pathogenetic loop system.

We are aware that, although our loops cover major etiopathological mechanisms of ALS, other mechanisms have been suggested to play a role in pathogenesis, such as deficits in axonal transport [7], endoplasmic reticulum stress, and proteostasis [57]. However, in mutant forms of the disease the two latter points are correlated to misfolded protein accumulation within mitochondria, thus being, to some extent, embodied in our model. Apart from speculation whether our model is exhaustive or not, the significant result of our approach is having arranged a wide set of data into a system of loops whose dynamics can account for the transition that generates the disease. Hence, the loop systems can be used as a dynamic map for the identification of druggable targets whose treatment could be allegedly able to block this transition.

As shown by the above structural influence matrix (12), the elements of our loop system can be divided into two synergistic groups that either promote (AMPK, GLUTAMATE, PFK3) or prevent (OXPHOS, PPP, APC$^{Cdh1}$) the pathogenic status. Such a result provides a clear indication for the development of possible therapeutic treatments. However, the interaction involving PFK3 and the PPP pathway is the most critical, as its deletion would disrupt multistationarity in the system, suggesting that its targeting could be sufficient to prevent the transition from physiological to pathological condition. Therefore, according to this model, the best druggable targets should be sought inside the complex of mechanisms that modulate PFK3 levels within MNs, aimed at maintaining low cellular level and activity of this enzyme system. A validation of this assumption could be obtained by exploiting *in-vitro*, *ex-vivo*, or even *in-vivo* models from SOD1$^{G93A}$ mice, using specific readouts downstream the target after manipulations aimed at reducing the PFK3 levels. The envisaged crucial role of PFK3 could also explain the lifetime of disease insurgence. As said above, PFK3 is maintained at low levels in neurons due to targeting to proteasomal degradation, which in our model represents a leakage out of the loop system. This is likely to slow down the transition of the system that gives rise to neurodegeneration and becomes eventually manifest at the symptomatic phase of the disease.

Finally, an added value of the proposed mathematical system is the possibility of extending our approach to other CNS pathologies, like Alzheimer or Parkinson's diseases that share

various neurotoxic mechanisms with ALS, including mitochondrial damage, oxidative stress, and excitotoxicity [58, 59]. In this respect, it is notable that the role of a positive feedback loop has been envisaged for Alzheimer [60], while elements of our loop system, like hyper-glycolysis and mitochondrial dysfunction, are also part of a post-concussion, neurometabolic dysfunction model [61]. This suggests that our loop model, possibly with suitable modifications, could profitably indicate new druggable targets for neurodegenerative and neurometabolic diseases, or that different diseases could share the same targets.

## Author Contributions

**Conceptualization:** Bruno Burlando.

**Formal analysis:** Giulia Giordano, Franco Blanchini.

**Investigation:** Marco Milanese, Tiziana Bonifacino, Silvia Ravera.

**Supervision:** Giambattista Bonanno.

**Writing – original draft:** Bruno Burlando.

**Writing – review & editing:** Bruno Burlando, Marco Milanese, Giulia Giordano, Tiziana Bonifacino, Silvia Ravera, Franco Blanchini, Giambattista Bonanno.

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
