## [Decision Letter · Decision Letter 0]

22 Oct 2020

PONE-D-20-26823

Modeling the pathogenesis of amyotrophic lateral sclerosis by a multistationary loop system involving mitochondria and glucose metabolism

PLOS ONE

Dear Dr. Giordano,

Thank you for submitting your manuscript to PLOS ONE. After careful consideration, we feel that it has merit but does not fully meet PLOS ONE’s publication criteria as it currently stands. Therefore, we invite you to submit a revised version of the manuscript that addresses the points raised by our expert reviewer.

We look forward to receiving your revised manuscript.

Kind regards,

Renping Zhou

Academic Editor

PLOS ONE

Journal Requirements:

Reviewers' comments:

Reviewer's Responses to Questions

**Comments to the Author**

1. Is the manuscript technically sound, and do the data support the conclusions?

Reviewer #1: Yes

2. Has the statistical analysis been performed appropriately and rigorously? 

Reviewer #1: Yes

3. Have the authors made all data underlying the findings in their manuscript fully available?

Reviewer #1: Yes

4. Is the manuscript presented in an intelligible fashion and written in standard English?

Reviewer #1: Yes

5. Review Comments to the Author

Reviewer #1: The paper titled "Modeling the pathogenesis of amyotrophic lateral sclerosis by a multistationary loop

system involving mitochondria and glucose metabolism" presents a systems engineering/control theory analysis of three interlinked feedback loops involving the mitochondria of spinal motor neuron cells during Amyotrophic lateral sclerosis (ALS).

The manuscript presents timely and important work, novel and intriguing results. However, the organization, writing and presentation make these results hard to understand and will limit the impact of this work. I believe it can be substantially improved, and I suggest the paper be accepted after a rewrite.

1. Reorganization

- The Methods section is short, and it is confusing as to what the is trying to be said here.

- The definition of the model is fragmented and spread throughout the result section

- There is no clear division between how the model was constructed and the analysis of the model.

I suggest that the authors remove the "methods" section and replace it with a "model" section. Move to this new section (including subsections) all detail about the modeling methodology, how the model was developed, and details on the variable, interactions, and feedback loops. It is rare that I want more equations in a paper, but I feel that a few more might bring clarity to the model description.

In the "results" section, clearly describe each "computational experiment" (or analysis) you performed on the model, what the results of the analysis showed and what the implications are. I feel that more organization here will help show the power of your results.

2. Rewrite abstract

- The abstract is very dense, and does not read either clearly or informatively. I suggest rewrite/workshop/wordsmith the abstract. I think that a shorter, more sufficient abstract will increase the readability, and thus the impact of this paper.

3. Consider changing the title

- I am always hesitant to recommend authors to change their title, however some revision here could help readers understand the content of the paper and increase the reach and impact of the publication. Informing the reader of what results you have in the title can be helpful.

One possible suggestion would be: "Modeling of ALS feedback loops elucidate critical molecular interactions".

6. PLOS authors have the option to publish the peer review history of their article (what does this mean?). If published, this will include your full peer review and any attached files.

Reviewer #1: **Yes: **Brian J Drawert

---

## [Author Response · Author response to Decision Letter 0]

16 Nov 2020

Response to Reviewer #1

We thank the Reviewer for the time he devoted to the review of our manuscript and for his very positive assessment.

We have thoroughly and carefully addressed all the received comments, changing the organization, writing and presentation as the Reviewer suggested.

We believe that the revised manuscript is now indeed substantially improved.

1. Reorganization

We have completely reorganized the manuscript following the Reviewer’s suggestions. The manuscript now includes a section (“The Model”) that discusses in detail the modeling and methodology (both biological modeling of ALS pathogenic mechanism, and mathematical model and methods). This section has been further subdivided into three subsections, i.e. a Background subsection containing the assumptions for the development of the model, a subsection devoted to the definition of loops, and finally a subsection describing the mathematical approach. Thereafter, the Results section has been devoted to describing the mathematical analysis performed on the model, its results, and the ensuing implications.

2. Rewrite abstract

As the Reviewer suggests, we have completely rewritten the abstract in a streamlined and more readable form. Most acronyms have been omitted in the effort of achieving better readability.

3. Consider changing the title

We have replaced the title with a more informative one by following and rearranging the Reviewer’s suggestion.

Please see the enclosed "Response to Reviewers" pdf file for further details.

---

## [Decision Letter · Decision Letter 1]

7 Dec 2020

A multistationary loop model of ALS unveils critical molecular interactions involving mitochondria and glucose metabolism

PONE-D-20-26823R1

Dear Dr. Giordano,

We’re pleased to inform you that your manuscript has been judged scientifically suitable for publication and will be formally accepted for publication once it meets all outstanding technical requirements.

Kind regards,

Renping Zhou

Academic Editor

PLOS ONE

Additional Editor Comments (optional):

Reviewers' comments:

Reviewer's Responses to Questions

**Comments to the Author**

1. If the authors have adequately addressed your comments raised in a previous round of review and you feel that this manuscript is now acceptable for publication, you may indicate that here to bypass the “Comments to the Author” section, enter your conflict of interest statement in the “Confidential to Editor” section, and submit your "Accept" recommendation.

Reviewer #1: All comments have been addressed

2. Is the manuscript technically sound, and do the data support the conclusions?

Reviewer #1: Yes

3. Has the statistical analysis been performed appropriately and rigorously? 

Reviewer #1: Yes

4. Have the authors made all data underlying the findings in their manuscript fully available?

Reviewer #1: Yes

5. Is the manuscript presented in an intelligible fashion and written in standard English?

Reviewer #1: Yes

6. Review Comments to the Author

Reviewer #1: (No Response)

7. PLOS authors have the option to publish the peer review history of their article (what does this mean?). If published, this will include your full peer review and any attached files.

Reviewer #1: **Yes: **Brian J Drawert

---

## [Editor Report · Acceptance letter]

9 Dec 2020

PONE-D-20-26823R1 

A multistationary loop model of ALS unveils critical molecular interactions involving mitochondria and glucose metabolism 

Dear Dr. Giordano:

I'm pleased to inform you that your manuscript has been deemed suitable for publication in PLOS ONE. Congratulations! Your manuscript is now with our production department. 

Kind regards, 

on behalf of

Dr. Renping Zhou 

Academic Editor

PLOS ONE